# Understanding the Use of Bio-Inspired Design Tools by Industry Professionals

**DOI:** 10.3390/biomimetics7020063

**Published:** 2022-05-18

**Authors:** Noah Pentelovitch, Jacquelyn K. Nagel

**Affiliations:** 1Research and Advanced Development Division, Helen of Troy—Home and Outdoors, New York, NY 10001, USA; 2Department of Engineering, James Madison University, Harrisonburg, VA 22807, USA; nageljk@jmu.edu

**Keywords:** bio-inspired design, biomimicry, design, design study, tools, industry, focus group

## Abstract

Bio-inspired design (BID) has the potential to evolve the way engineers and designers solve problems. Several tools have been developed to assist one or multiple phases of the BID process. These tools, typically studied individually and through the performance of college students, have yielded interesting results for increasing the novelty of solutions. However, not much is known about the likelihood of the tools being integrated into the design and development process of established companies. The mixed-methods study presented in this paper seeks to address this gap by providing industry engineers and designers hands-on training with the BID process and four BID tools. Understanding which tools are valued and could be adopted in an industry context is the goal. The results indicate multiple encouraging outcomes including that industry practitioners highly valued the process framework tool (BID canvas) as it allows for flexibility in tool use, as well as valued learning with a suite of BID tools rather than a single one to accommodate different workflows and ways of thinking.

## 1. Introduction

Bio-inspired design (BID) has served as an inspiration and motivation for many engineers and designers to explore the natural world to find novel solutions to problems. Many BID tools and methods have been researched and written about [1,2], and many examples have been lauded in popular media [3,4,5,6,7].

Throughout history, insights from one field, carried over to another, have driven breakthroughs and discoveries that have led to innovations [8]. Flora and fauna such as beaver dams, bird wings, and gecko feet have inspired novel solutions. The process for arriving at these solutions is often framed as a “eureka!” moment or the result of decades of lab work. It is difficult for engineers and designers to systematically apply knowledge from the field of biology to problems in their industries, despite headlines loudly proclaiming that nature has the answers. Tools have been developed to assist in jumping over the hurdle of finding and applying biological inspiration. However, none have emerged as an obvious solution to bridging the gap between engineering and biology.

Human-centered design (HCD) and “design thinking” have become standard practice in consumer-product design and development. Initially, these were areas of primary concern to industrial designers, but as the value of HCD became apparent across industries, it began to be integrated into engineering and business school curriculums. Now, it is common for industrial designers, engineers, and product managers to be conversant in the process and methodologies of HCD. Using similar language and working from a shared base of knowledge leads to clearer and faster communication and better comprehension of the motivation for and importance of design features. Innovation is achieved through products that better fit the needs of end users.

The next evolution in transdisciplinary collaboration for design and engineering may very well come from biology. Nature has a wide variety of solutions to functional problems that can be drawn upon for inspiration. However, this corpus of knowledge is of a different composition than what designers and engineers are familiar with [9]. To utilize knowledge from biology, engineers and designers must first be able to comprehend it, then translate it into a context that is relevant to the problem they are solving. One way to do this would be to have a biologist as part of the team [10,11]. Another way is to introduce tools and processes for practicing BID.

In industry, more experienced engineers and designers tend to set the standard for processes and practices. As new engineers and designers join a company, they are mentored by those with more experience, who pass on their processes and practices, thus solidifying a culture and set of expectations for how new ideas are generated and tested. This creates a significant barrier to introducing a completely new approach to problem solving in industry, such as bio-inspired design. For a new approach to be adopted, it needs to be comprehensible and relevant to the engineers and designers who will practice it. It also needs to be supported at the highest levels of the company to justify the time it will take to learn a new way of doing things. BID presents an additional complexity in that it is unlikely that any company involved in engineering and design will have a biologist on staff, leaving engineers and designers to attempt the process without the benefit of an expert. Thus, engineers and designers that want to apply BID will need to leverage existing tools and processes. Although there are a growing number of publications on tools and processes for practicing BID, challenges to adoption remain, such as (1) the lack of addressing the inherent challenges in organizations; and (2) industry engineers and designers being infrequently engaged in the studies to determine how, or more importantly, if, the tools and processes could be adopted.

The research carried out on the efficacy of tools for systematically applying BID has typically been performed with college students [12,13,14,15,16,17,18,19,20,21,22,23,24,25,26,27,28]. While college students provide large, easily accessible sample sizes, they do not have the same level of work experience, nor do they work within similar constraints as industry engineers and designers. For example, industry engineers and designers are constrained by the specific business needs of their organization as well as the materials and manufacturing processes common to their field. Development timelines and budgets are also primary drivers of which solutions can be applied to problems. Furthermore, the culture and expertise within an organization drive how likely a tool is to be adopted. Thus, to know if a BID tool could be successfully put into practice, more research with industry engineers and designers is required.

The study presented in this paper seeks to address this gap by providing industry engineers and designers hands-on training with the BID process and four BID tools. The primary goal of the study is to understand which tools are valued in an industry context, which could give insight into which could be adopted. Two consumer goods brands, OXO and Hydro Flask, were involved in the study. These brands represent design leaders in their respective product categories; however, it is important to note that no one company operates the same as another, and what works for engineers and designers in one company may not work for another.

## 2. Background

Various tools and processes have been well studied for performing bio-inspired design. These tools are generally studied to assess their effectiveness at generating novel solutions rather than for their long-term adoption. Methods such as Bio-TRIZ [29], PeTAL [30], biological functional modeling [31], E2B Thesaurus [32], the Aalborg BID method [33], the biomimicry design method [34], the spiral design method [35], the bio-inspired design method [21], and the bio-solution in search of a problem method [21] have been developed to facilitate the translation of principles from nature to usable solutions for engineering, design, and business problems.

Previous research on BID tools has focused on the value of a tool or method for achieving an explicit end result (i.e., “more sustainable” or “more creative”). Mead et al. evaluated the relative improvement in the sustainability of solutions when using bio-inspired design [36,37]. The study found that companies that use BID as a long-term approach to innovation found greater success with sustainability-oriented innovation. Kennedy et al. compared the use of far-field biological vs. industrial analogies during concept generation with industry participants [38]. This study only examined whether the novelty/creativity of concepts increased. It did not assess the likelihood of this tool being used or whether it generated practical and usable ideas. Kennedy et al. [39] also explored frames of inquiry with industry professionals for the beginning or exploratory part of the BID process. The study focused on investigating whether frames of inquiry could lead to finding a larger number and a wider variety of inspiring biological models. The study did not find significant positive effects. This study is especially relevant as it was carried out with industry professionals and calls attention to the gap between the desire of R&D professionals to use biomimicry and the lack of definitive industry-based studies on which to base best practices.

It seems logical to assume that a demonstrably better tool or process should justify adoption on its own; however, the reality is that companies develop over time to reinforce existing processes by hiring people whose skills fit into the business needs and culture. New tools, and especially new processes, require a company to disrupt what they have built themselves up around. A new process slows an experienced employee down and disrupts team workflows, something few companies are willing to do as they are incentivized to stay on schedule and within cost, no matter how valuable the change could be. Barriers to implementing BID in product development and BID product commercialization have been researched, but do not examine the influence of individual tools in industry applications [40,41].

There is little written on teaching multiple tools for the purpose of understanding what engineers and designers would be most likely to adopt. Faludi et al. studied three design approaches to understand how the components of the approaches were perceived by industry engineers, designers, sustainability professionals, and managers for value in two categories: sustainability and innovation [42]. The study revealed that some components of each design method were favored more than others and that the participants viewed them not as requisite parts of a process but as individual tools that they could use in isolation outside of the prescribed process. This reflects both a desire and a need amongst industry professionals to have a hybrid approach to their design process, custom built to meet their own needs and preferences.

There is evidence that industry has adopted some BID practices, though it is unclear whether this has led to repeated or planned use of BID for subsequent problem solving. There are prominent examples of commercialized bio-inspired solutions, including: Whale Power [43], Geckskin [44], Cora Ball [45], Interface Carpets [46,47], PAX [46,48], Encycle [46,49], Skarklet [46,50], VELCRO [51], and Japanese bullet trains [52,53]. A few shoe companies (Nike (Beaverton, OR, USA), New Balance (Boston, MA, USA), and ASICS (Port Island, Kobe, Japan)) found inspiration from animals for shoes [46,54]; however, it is not clear from new product offerings or available information whether this continues to be put into practice.

There is an important difference between companies that have used bio-inspired design to develop a product (Interface Carpets (Atlanta, GA, USA), PAX (San Rafael, CA, USA), Cora Ball (Middlebury, VT, USA)) and those that took research and spun it out into a company centered around the innovation (Whale Power (Toronto, ON, Canada), Geckskin (Somerville, MA, USA), Sharklet (Aurora, CO, USA)). An established company has more hurdles to trying new approaches to problem-solving than a start-up: legacy knowledge, established processes, cost constraints, material and manufacturing constraints, and assumptions about what innovation should look like. Bio-inspired design requires that engineers and designers engage in a time-intensive, unfamiliar process with subject matter that may be difficult to grasp. There is a desire within companies to utilize time efficiently and towards company priorities. Bio-inspired design researchers do not necessarily take into account the constraints on industry engineers and designers and consider what they need to integrate BID tools into their existing workflow.

## 3. Materials and Methods

The research question motivating this work is: what is the value of BID tools in an industry context? To assess the value of BID tools in an industry context, a mixed-methods study was created and is explained in the following subsections.

### 3.1. Study Design

Engineering and design professionals from OXO and Hydro Flask, two consumer-goods brands in the Home and Outdoors division of Helen of Troy, were invited to participate in training centered on learning bio-inspired design and four selected tools. OXO makes high-quality houseware products for everything from cooking to cleaning to children’s products. Hydro Flask makes outdoor products for keeping drinks and food hot or cold while on the go.

Participation in the training occurred virtually through MS Teams over the span of five weeks. Continuing professional-development training is considered an activity within the scope of regular job duties; thus, no compensation was provided for participation. The training included three major parts: (1) learning and applying a fundamental BID process; (2) learning, applying, and self-reporting on the use of four different BID tools (quantitative); and (3) discussion of the experience (qualitative).

In week one of the training, the participants received a one-hour presentation on the BID process (as shown in Figure 1) with examples followed by a one-hour application session that resulted in baseline data. An industry-relevant design challenge provided by OXO was used for the application session. The challenge was selected as it addresses a number of critical functional challenges faced by current OXO products. The challenge is as follows:


*Soap-/oil-/dressing-/sauce-dispensing devices are convenient and popular but have many drawbacks: pumping mechanisms are difficult to clean, require high pumping force to move viscous fluids, and include many small parts and a difficult assembly. Nozzles can easily become clogged with viscous or grainy fluids. The volume of dispensing is fixed or hard to adjust. Redesign a soap-/oil-/dressing-/sauce-dispensing device to address one, some, or all of the above issues.*


In week two of the training, the participants received a two-hour presentation covering the four BID tools used in the training. The tools are listed and described in Table 1. Four tools were chosen to ensure the breadth of tool exposure and assistance across the entire BID process. The BID tool selection was based on the second authors’ involvement in their creation, thus leading to a high level of confidence for training others in how to use them. All presentations were recorded and accessible through the MS Teams platform for future reference.

After tool introduction, it was expected that each participant would spend approximately two hours applying each BID tool to the same OXO design challenge and completing a feedback form. The order of BID tool application was randomized for each participant to control for participants generally getting better at solving the same design prompt. During weeks two, three, and four of the training, two-hour weekly working sessions were held to assist participants with blocking off time in their week to complete the training tasks. In week five, participants met virtually to discuss the experience and perceived value of the tools. An overview of the five-week training is given in Figure 2.

### 3.2. Participants

This study involved a total of 11 participants from management, engineering, and design that work full-time for one of two consumer-goods brands. Table 2 provides a breakdown of the participants’ roles within the brands as well as the collective credentials represented by those individuals. The majority of participants had worked in their current role for at least one year with the longest length of time being four years. Across their careers, however, the number of years spent involved in design or product-development work spanned from 1 to 10 years.

The participants shared a number of important similarities. All participants worked at a design-oriented consumer-product company under the same management and within the same process. All participants have collaborated with one another on projects and worked within the same material, manufacturing, and cost constraints prior to this study. Similar language, tools, and problem-solving approaches were utilized or, at least, known to one another.

All participants worked in or with the Product Development Team at OXO and Hydro Flask. At OXO and Hydro Flask, industrial designers, product engineers, and design engineers work together to develop and bring products to market. Industrial designers and design engineers focus on the front half of development while product engineers are involved in a product’s development from concept through production. The Advanced Development Team (first row of Table 2) supports and collaborates with the Product Development Team (second through fourth rows of Table 2) by carrying out early-stage investigation of materials and manufacturing methods as well as contributing domain expertise when appropriate. While there is significant collaboration and overlap between participants in their roles, there are specific differences in their primary job function and education. Industrial designers are primarily focused on the visual and functional design of products. Design engineers, compared to product engineers, work more closely with industrial designers to put concepts into CAD, prototype, and brainstorm concepts. Product engineers collaborate with design engineers and industrial designers but work more closely with manufacturers to assure that products can be manufactured reliably, the right material is selected, and the product will function as designed. The Associate Directors of Advanced Development act in an R&D capacity, focusing on upstream innovation and new solutions in materials and manufacturing. Operations engineers are primarily focused on ongoing improvements to existing products and interact most often with factories.

Half of the participants (50%) had read or watched videos about bio-inspired design but had never tried it. The minority of participants (20%) were familiar and had tried to apply it once or twice. These prior experiences tended to be while the participant was still in school and before working as a professional. The remaining participants (30%) had no experience with BID.

### 3.3. Data Collection

Data were collected at four points in the training: (1) pre-survey, (2) after learning the BID process, (3) after using each of the four BID tools, and (4) a post-training discussion of the experience.

The pre-survey questions focused on the professional history of the participants and included information regarding professional credentials and titles, the length of experience in their current role and design or product-development work, level of experience with bio-inspired design, typical workflow for design or product development work, and expectations of the training. This information was collected prior to the start of the training.

In week one of the training, each participant was given one hour to apply what they had learned about the BID process fundamentals to an industry-relevant design prompt. These data were collected digitally via MS Teams. Each file contained information for each of the five steps of the BID process as shown in Figure 1. These data are considered the baseline data as the participants had not been exposed to the four BID tools yet.

During the three weeks of BID tool application, a slide-deck-based feedback file was used to collect quantitative responses on eight questions, a qualitative response to a process question and a place to include the bio-inspired solution that resulted from using the tool. The feedback file was completed following each BID tool for a total of four files for each participant.

The following quantitative questions were provided with a scale of 1—strongly disagree, 2—disagree, 3—agree, and 4—strongly agree on the second slide of the feedback file:Rate your value of the tool you used.The tool effectively helped you work toward a BID solution.The tool improved your confidence about doing BID.The tool effectively helped you build connections between biology and the problem.The tool was intuitive enough to use on your own.The tool reduced your need/reliance on a biologist.Using the tool resulted in a better outcome than the baseline case (no BID tools).You would use the tool again.

The qualitative process question asked participants to move the five provided icons to the part of the BID process graphic (Figure 1) where the tool was used. Participants were instructed to place at least one icon, but up to five. To capture the output of using the BID tools, the final slide instructed participants to “Please use as many slides as needed to capture your work as a result of using the BID tool”. Photos and scans of handwritten or drawn work were placed there. Each participant was given time weekly by management to work on the training tasks and better their understanding of bio-inspired design. Of the eleven participants, eight (73%) completed all tasks.

The post-training discussion was conducted as a focus group consisting of those that participated in the training. A focus group allows for the emergence of a range of ideas as well as variation in perspectives between individuals, which is critical to assessing the value of the tools included in the training. Focus groups are used in multiple engineering research areas, including software engineering [59], requirements engineering [60], and design [61,62]. A facilitator provided the topical or question focus of the discussion, and the data collected came from individual responses and group interaction that occurred during the discussion [63]. The discussion occurred as a video conference in the MS Teams platform and was recorded. At the conclusion of the post-training discussion, each participant was asked to share which BID concept created they were most confident about or was their favorite.

### 3.4. Data Analysis

Quantitative and qualitative data analysis was performed. Quantitative data from the eight questions in the feedback forms were averaged across participants and summed. Due to the limited number of participants, statistical analysis was not possible. Data from the qualitative process question were tallied to look for unexpected tool use with respect to the BID process. The collected BID concepts were not analyzed due to the focus being on perceived value of the BID tools. The number of self-selected concepts per BID tool, however, were counted.

Qualitative data from the post-training discussion were extracted from the video recording. Participant responses were transcribed into text that included timestamps and observational data of group interactions including head nodding and verbal agreement. Thematic analysis was used to analyze each topic and question [64]. Concept maps with line thicknesses that denote the importance of each theme were created to visualize the results of the analysis.

## 4. Results

The order of the data analysis results presented in this section begins with quantitative and ends with qualitative. First, the feedback from results for all four tools are presented, followed by participant reported concept selection with the associated tool used to create it. Tool process use is presented third, with focus-group qualitative analysis presented fourth.

Table 3 summarizes the results of the eight questions in the feedback forms organized by the BID tool. The results are ordered from the highest to lowest total mean value score. The visual analogy sketching technique received the highest scores overall in the feedback forms. It is the only tool to receive mean scores of three or higher across all eight questions, as well as a mode value of four for all but one question. At the other end of the spectrum is the BioSearch tool with all mean scores below three and the only tool to have a bimodal distribution for one of the questions. BID Canvas and the E2B Thesaurus were close in the total mean score but have differences when it comes to intuitiveness, improving confidence, and future use.

A similar trend is found with participants reported concept selection. Of the eight participants that completed all tasks, five selected a concept created using the visual analogy sketching tool and three selected a concept created using the BID Canvas tool as the one they are most confident about. Zero participants selected a concept created with the E2B Thesaurus or BioSearch tools.

The results of the qualitative process question are given in Table 4. The maximum number of reported uses for each tool per BID process step is eight. The quantity of participants that self-reported using the BID tool in each step is indicated after each BID process step in both the expected and unexpected use columns.

The post-training discussion was facilitated as a focus-group discussion. The opening statements by the facilitator shared the purpose, topics, and questions of the discussion and allowed the participants to begin sharing their thoughts on any of the four BID tools. Participants spontaneously discussed the BID Canvas tool first and discussed it for the longest time of the four tools. Themes from this portion of the discussion are given in Figure 3. Line thicknesses denote the importance of each theme by how often it was represented in the discussion. *Provides a process flow* was the strongest theme. The following two participant statements that received acknowledgement from others in the group and increased the importance of this theme are:“I thought it worked really well as a framing device to figure things out and set a flow for how to get somewhere”.“It gives you place to go forwards and go backwards. There is a very clear process path”.

Sub-themes that emerged related to *provides a process flow* included the recognition that multiple ideas or avenues are identified, that more than one process flow is provided, and the benefit of putting biology and engineering in separate boxes and understanding them first before starting to put them together. A weak theme that was touched upon in conjunction with the strongest theme was *facilitates using multiple tools together*. Participants shared that the BID Canvas worked really well with the other BID tools and provided a way to bring them all together.

Participants discussed the structure of tools and how, once the structure is understood, the designer can move away from it and establish an individual process similar to using training wheels, which resulted in the *structured learning to develop individual process* theme. The next strongest theme was *found the tool useful*, which participants commented directly about and others supported through acknowledgements. At this point, the discussion shifted to include the Visual Analogy Sketching tool when a participant mentioned that the two tools they used the most are the BID Canvas and Visual Analogy Sketching. During the blended discussion, one clear theme emerged for each of the two tools. The strongest theme for Visual Analogy Sketching of *intuitive because sketching is part of current workflow* was one of the two, as shown in Figure 4. Many participants self-identified as being visual people and using sketching in their work. The following three participant statements that received acknowledgement from others in the group and increased the importance of this theme are:“This is generally how I do things anyway so I had more success with that”.“My workflow is heavily based on sketching anyway. I sketch out my engineering thoughts anyway so sketching biology was not different from what I already do”.“The visual analogy sketching was my comfort zone”.

Following the paise for the Visual Analogy Sketching tool, an alternative perspective was shared when a participant said “The canvas is good because I can get lost when I am just using visual analogy sketching”. This statement caused the BID Canvas theme of *knowledge space is critical to the process flow* to emerge. The discussion included how the knowledge space anchors the BID process by ensuring information is not overlooked, that the biological analysis is relevant, and that the process is followed better. The discussion then shifted toward the general topic of process and how product-design work happens. Connections between BID and current engineering processes were made that bridged back to Visual Analogy Sketching and resulted in the other three themes. The participants explained that sketching resulted in the identification of engineering principles (i.e., oleophobic) and exploration of how to achieve the principle using solutions from outside their industry, in addition to translating the biological system into a mechanical system which made it easier to understand.

Discussion spontaneously shifted toward the Engineering-to-Biology (E2B) Thesaurus when a participant commented: “I wish I had the E2B first because there were so many terms I hadn’t thought about or used. These tools are important to use as a whole rather than individually”. This tool had the shortest discussion time of the four and resulted in three equally weighted themes, as shown in Figure 5. Participants shared a mixture of positive and negative individual experiences of using the tool, as evidenced in the following statements:“I was expecting it to be difficult to use, but it allowed me to expand my search terms that I didn’t even think about and it was really useful”.“I found it interesting that I found myself typing keywords into search engines and finding things and learning, but then having to pull myself back because the exploration wasn’t relevant”.“I had a paragraph to translate with the E2B and it was tough. I didn’t really know what words to substitute. If I translated too many I lost the meaning, not enough and I couldn’t understand. It was hard to know what the right balance was”.

No participant voluntarily spoke up about the BioSearch tool. Discussion started when the facilitator asked the group about it. This tool had the second-longest discussion time of the four and resulted in one heavily weighted theme and three lesser-weighted themes, as shown in Figure 6. Participants shared mostly negative individual experiences of using the tool. The following statements relate to the themes of *too limiting in content* and *lacking desirable features*:“I felt like I wanted to use it or like it more, but in the current state it was too limited. It was too simple to find a jumping off point. I wanted a bigger library”.“I didn’t use it much, but when I did it made me want the textbook. I wanted to look at all the info around it”.“I wish the Nature4Innovation site and the BioSearch tool be smashed together. They are both broken in different ways”.

Following the negative comments, one participant shared an alternative perspective by saying “I am more receptive to the BioSearch because I am a verbal person. It is the fastest tool. Type a few words or phrases to get things moving”. This shifted the discussion toward positive comments, which resulted in the two lowest-weighted themes of *bridge to detailed information* and *text-only approach resulted in ideas that other tools did not*. The following statements relate to the more-positive themes:“I found terms in the E2B, then used those to search the BioSearch docs, found the topic I wanted to explore further, then went on-line. It was a good stepping stone”.“This was the most challenging tool for me. The results were in words so I had to think more abstractly because visuals might allow my mental models to start filling in the gaps. I think it challenged my visual way of thinking and that lead to some ideas I had not considered”.

Six other topical areas occurred during the discussion, and include: tool combination, when to apply BID in industry, BID light, the design process, talk to biologist, and expectations. The topic of design process, or how product-design work happens in industry, accumulated the most discussion time as it spontaneously occurred between other topics. Tool combination and expectations had near-equal discussion times and almost as long as the BID Canvas discussion. Talk to biologist was the next highest, followed by when to apply BID in industry. Of the non-tool discussion topics, BID light was the shortest but still longer than the E2B discussion.

When asked if it would have been beneficial to talk with a biologist, some participants agreed it would have been useful for obtaining guidance on which biological systems to learn about for inspiration, and an equal number thought that once they had the right direction for inspiration, they were confident in their ability to abstract biological information and apply it. The following statement, which was acknowledged by another participant in their own words, helps to make this clear: “It would be useful to talk to a biologist to get started in a particular direction. I tell them viscous fluids, and they tell me look at organism X, Y, and Z. But didn’t need them to make the abstractions”. All themes for this topic are shown in Figure 7.

When asked about when to apply BID in industry considering market-ready vs. research-and-development timelines, the discussion resulted in two equally weighted themes that occurred in serial order (top to bottom), as shown in Figure 8. Initially, participants discussed this question more abstractly by considering how problem framing can greatly influence the application. An example participant statement that is representative of the discussion and first theme, as well as received acknowledgement from others in the group is “Depending on the level of fidelity you are operating at, it can be very valuable. System level vs. component level”. The first theme was followed by multiple specific examples of how it might be applied in existing contexts such as manufacturing issues or addressing customer issues through incremental improvements, which resulted in the second theme. An example participant statement that is representative of the discussion and second theme is the following: “We have a product that is in the process of being made, but we find a hiccup that needs a solution. This is an opportunity to apply BID. This is a well contained problem space that makes it easy to apply”.

Although tool combination was spontaneously mentioned during the BID Canvas and E2B discussions, the participants were asked about the potential for combining tools later in the discussion. One participant answered with the statement: “I think it is good to know the structure and how to bend the rules. There is not a right way. Know the strength of each tool”. The discussion then focused on the value of learning tools and what was learned in college, followed by an acknowledgement that using a process or tool takes intention. Although this topic did not result in any meaningful themes, it did result in a participant spontaneously asking the group: “What is the BID light version that can be used every day or every week? The full version is needed when fresh ideas are needed. But is there is a light version that gives me some new inspiration and go from there?” Participants quickly reached consensus that a light version of BID would be to pull inspiration from intuitive knowledge, or what you already know, which was the only theme for this topic. Learning about biology on a regular basis can build a mental database or solution bank of things to consider when solving problems. One participant summarized the single theme and received acknowledgement from others when they said “This makes sense. Because when we design something, we often refer back to what we know. Therefore, if we can see more [biology], then we have a higher chance of applying a BID approach”.

Throughout the focus-group discussion, the topic of the design process kept coming up. The participants made comparisons and connections to how product-design work happens in their industry, which resulted in the six themes shown in Figure 9. The strongest theme of *industry-experience-biased solutions toward practical and feasible solutions* correlates to the participants using their multiple years of industry experience to objectively analyze and eliminate biological inspiration that would result in a novel idea that was not achievable. The following two participant statements that received acknowledgement from others in the group and increased the importance of this theme are:“We are likely doing some filtering of ideas to be applicable to what we can achieve. For example, I had a really hard time of ever getting away from the idea that a user will not push down on a soap dispenser”.“Good point about using BID as an inspiration, but limiting it to what is possible or necessary. It has to speak soap dispenser!”

The themes of *connection to current benchmarking practices* and *manufacturability and associated costs* had few comments but had a strong response from others with head nodding and verbal agreement, which increased the importance of these themes. Participants recognized that their design process typically starts with benchmarking products and made the connection that BID is similar as it also provides an opportunity to look at what has been achieved in nature and use that as inspiration for a new direction. With respect to manufacturing and costs, participants discussed how an idea might be made and how much it would cost.

Participants discussed that when searching for and learning about biological systems, it helped them to better understand the problem, which led to iterating or reframing the problem and resulted in the theme of *problem statement iteration*. The equally weighted theme of *design timeline constraints* was related to the fact that depending on the goal of the design task, whether BID or not, the timeline can be drastically different. Participants also recognized that learning about a biological system led them to an idea that was already on the market but used in other fields; thus, it was feasible, and bringing it to their industry would be innovative. This resulted in the last theme of *bridge to solutions in other fields*.

The final discussion topic of expectations was not analyzed; rather, participants that shared statements are provided in the following list:“Will I use the BID canvas once a month? Probably not. But do I think this was overall a useful, positive experience that has provided tools and a way of thinking about things more specifically than anything. Yeah, I absolutely do!”“It feels like another tool that I can pull out when needed. Which is a good thing. I learned more about how to make the connection between biological and engineering systems”.“For me this was a very valuable experience. BID has this mythology or mystery around it, but now I can frame it as biology is something I can go to the same way as aerospace or electronics or other industry and ask how did they approach it. ….. [The biology] is more accessible now. These tools made it so that you just needed something to put it in the right context for absorbing. These tools I do feel like help contextualize things better”.“Not a waste of time. I found it very useful”.“I wish I had more time to spend with the tools and in a more collaborative environment. It was also challenging to make the collaborative sessions because this was more about professional development than work with a deadline”.“I wanted to think more about biology and how to understand it. I took-away new resources for long-term inspiration and sketching to answer a question. The tools helped me to put the information into the right context”.“There was something nice about spreading this out over an extended period. Hopefully promotes better learning”.

## 5. Discussion

The results of the study show a preference among participants for some tools more than others but value in all of them. Visual Analogy Sketching and BID Canvas were the highest rated in the survey questions, had the most positive comments, and were the first two tools mentioned by participants in the discussion. The concepts created with these tools were also most preferred by the participants. The BioSearch tool received the lowest scores in the survey, but participants felt it had value within the process for its ease of use and speed of search results. Despite stronger negative themes overall for the BioSearch tool, comments such as “I felt like I wanted to use it or like it more, but in the current state it was too limited... I wanted a bigger library” demonstrate a recognition that the tool itself provides value and there was a desire not to abandon it, but to see it evolve and expand.

The E2B Thesaurus did not emerge as a highly preferred tool, but the quantitative analysis as well as some of the qualitative feedback during the discussion showed that participants recognized its potential. This was shown by the statements: “I was expecting it to be difficult to use, but it allowed me to expand my search terms that I didn’t even think about and it was really useful” and “I wish I had the E2B first because there were so many terms I hadn’t thought about or used”. A counterpoint to this was that when to apply the E2B Thesaurus and the rules for doing so were confusing and felt complicated. Participants mentioned utilizing the E2B Thesaurus when using the BioSearch tool, even though this was not discussed or encouraged during the training. The E2B Thesaurus is an example of a tool that, like BioSearch, may not be highly valued on its own as it is difficult to apply and does not fit into a familiar process. However, when it is combined with other tools, it can have value in supporting different parts of the BID process.

While all four tools were valued, Visual Analogy Sketching and the BID Canvas stood out as the most preferred. It is helpful to look at the themes that came up during discussion to better understand why this may have been. A key difference between the four tools becomes apparent when looking at Figure 3, Figure 4, Figure 5 and Figure 6. For the BID Canvas, three of the five themes that emerged during discussion were positive themes related to process or workflow. In Visual Analogy Sketching, there were two out of four. By comparison, E2B Thesaurus and BioSearch had no themes related to the tools fitting into a process or workflow. This difference is worth highlighting as the preference for Visual Analogy Sketching and BID Canvas may have as much, or more, to do with how they fit into the participants workflow as they do with the strength of the tools alone.

A potential explanation for the preference for Visual Analogy Sketching overall is that this is a tool that is already used outside of a BID context at both brands. Engineers and designers regularly use and are quite comfortable with sketching to communicate and refine ideas and both brands are design-oriented consumer-product brands. Visual Analogy Sketching was, therefore, an easy leap to make and would have been comfortable to use. This is expressed in the feedback “This is generally how I do things anyway so I had more success with that,” and “My workflow is heavily based on sketching anyway. I sketch out my engineering thoughts anyway so sketching biology was not different from what I already do”. The similarity of a BID tool to a tool already used by a designer or engineer, such as sketching for communicating ideas, likely contributes to an overall preference for one tool over another. This should be kept in mind while working with other professions or industries and an example of where teaching multiple tools has an advantage.

By comparison, the BID Canvas, while not difficult to understand, did require participants to follow a prescribed process rather than fitting into their existing one. It might be expected that a new process would be difficult for users to adopt, especially given the focus on personal process that participants expressed in the discussion. However, BID itself is a relatively new idea for the participants, especially being applied in an industrial setting. Participants found value in the framework provided by the BID Canvas and how it facilitated the BID process and that it already closely reflected their personal process. This can be seen in the emergence of the themes of “helpfulness” and “utilization of multiple tools” for the BID Canvas. Each of the four tools meets both unique and overlapping needs, and the BID Canvas provides a framework on top of which the user can integrate multiple tools.

Participants used all the tools in both expected and unexpected ways, as shown in Table 4. There were more associations with expected use across all tools, but participants showed a willingness to try each tool in almost all process steps. There was somewhat more unexpected usage of Visual Analogy Sketching and the BID Canvas tools, though the overall sample size is too small to draw any conclusions. A potential explanation for this is the comfort level participants felt with those tools, as reflected in the higher survey scores and overall preference for them. Another possible explanation is the pre-disposition of this group of professionals for visual and structured tools.

The participants in this study work in design-oriented brands where visual communication is heavily utilized. While a pre-disposition for visual communication is likely true for many engineers and designers at these two brands, it is also not true of all. This study showed how teaching multiple tools allows a group of professionals to select those tools that most align with the way they think: visual or verbal, structured or unstructured. Participants expressed the challenge of this, but also recognized the benefit. For example, when discussing the BioSearch Tool, one participant said: “This was the most challenging tool for me. The results were in words so I had to think more abstractly because visuals might allow my mental models to start filling in the gaps. I think it challenged my visual way of thinking and that lead to some ideas I had not considered”. By comparison, another participant said: “I am more receptive to the BioSearch because I am a verbal person”.

The results of this study show an interest from industry engineers in combining multiple tools when engaging in BID. This insight also emerged from a previous study on the use of multiple tools by industry professionals [39]. As participants learned the BID process, they were also considering how it would fit into their existing workflow and how each tool could best be utilized, something also seen by Rovalo et al. [65]. This was evident from the many times over the course of discussion that personal processes came up as a subject of conversation. Compared to students, an industry professional must juggle many more constraints, most importantly the necessity that a solution is practical, producible, and cost-effective. This idea was expressed during discussion: “We are likely doing some filtering of ideas to be applicable to what we can achieve”. This insight is important for understanding the value of the concepts that come out of a group of engineers and designers from industry versus a group of students.

If BID is to be successful at a company, it needs to produce ideas that can be used and ultimately be brought to market. Industry professionals want to adopt new tools that can help accomplish this goal but want flexibility to integrate tools in a way that works best for them. This sentiment was expressed by one participant when they said: “I think it is good to know the structure and how to bend the rules. There is not a right way. Know the strength of each tool”. Furthermore, the pre-survey question asking “How do you typically do design or product development work? Could you describe your typical workflow?” revealed that there is not a consistent workflow used across the participants; thus, they rely on an individually developed approach. Tools that afford flexibility lower the barrier to adoption.

The wide array of tools that have been developed for performing BID is a testament to the thorny nature of identifying and translating principles in biology to useful solutions. Teaching a suite of tools gave the participants the opportunity to envision how they could work with the tools individually and combined. The result was that participants felt much more comfortable with the BID process. One participant summarizes this well: *“For me this was a very valuable experience. BID has this mythology or mystery around it, but now I can frame it as biology is something I can go to the same way as aerospace or electronics or other industry and ask how did they approach it. ….. [The biology] is more accessible now. These tools made it so that you just needed something to put it in the right context for absorbing. These tools I do feel like help contextualize things better”.*

It is important to note that the participants in this study worked in the same industry and for sister brands. It should not be assumed that the specific feedback on the tools taught in this study are generally applicable to all engineers and designers. It is likely that the participants in this study were already pre-disposed through training, the industry they chose to work in, or their personal style to feel most comfortable communicating visually. Another group of professionals, in another industry or from a different educational background, may find more comfort in text-based or verbal tools. This is even more reason why BID tools should be studied as a suite rather than individually, as each group of industry professionals may find different aspects of each tool applicable to their specific needs.

A challenge and strength of this study was that it was carried out remotely. While participants were able to join from more locations and over a longer period, engagement with the sessions was difficult to gauge, and some participants had trouble joining for every session. Conducting the training and application sessions in person may have facilitated better engagement and understanding as questions could be asked in real time and participants would have stayed focused on the task at hand. However, the prolonged engagement over the span of four weeks was an authentic experience of integrating learning a new skill into daily work. A limitation of this study is the small sample size, which limits the quantitative analysis.

## 6. Conclusions

This study set out to explore the value and utility of BID tools from the perspective of practicing industry professionals, where the adoption of bio-inspired design as a problem-solving tool is desired but rarely achieved. Many tools and processes have been developed to facilitate inspiration from nature. However, these tools have mostly been researched with student populations and have used the output of a tool as the determinant of its value.

We propose that to increase the adoption of BID in industry, industry professionals must be the ones providing feedback on the value of a tool. A tool that cannot be readily adopted is not useful, and a tool cannot be adopted if the user does not understand how it fits into their work. The tools for facilitating bio-inspired design that focus on usability and integration with tools and processes already in use rather than optimizing for a specific output (i.e., novelty, sustainability, etc.) have a higher likelihood of adoption. We propose that teaching a suite of tools has higher value to industry professionals than teaching a single tool. The participants in this study found value in all four tools and expressed a desire to combine them in various ways for different parts of the BID process. This more readily reflects the nature of working in industry, as the process is less rigid, and individuals are allowed to utilize any tools or approaches that may accomplish their project goals.

When introducing a new tool to a group of engineers, designers, or other professionals, it is best to explore how it might fit into their existing process. The participants in this study frequently referred to their own process and how the tools that they learned fit into it. Each engineer or designer has developed their own process that has been refined over time to best suit their personal skillset, project requirements, and company expectations. For these reasons, it is unsurprising to see that the most preferred tools were also the easiest to comprehend and integrate into existing workflows. Tools that give the user flexibility in how and where they are used in an engineer’s or designer’s process will lower the barriers to their adoption, increasing the likelihood that they become a regular part of their problem-solving toolbox.

## Figures and Tables

**Figure 1 biomimetics-07-00063-f001:**
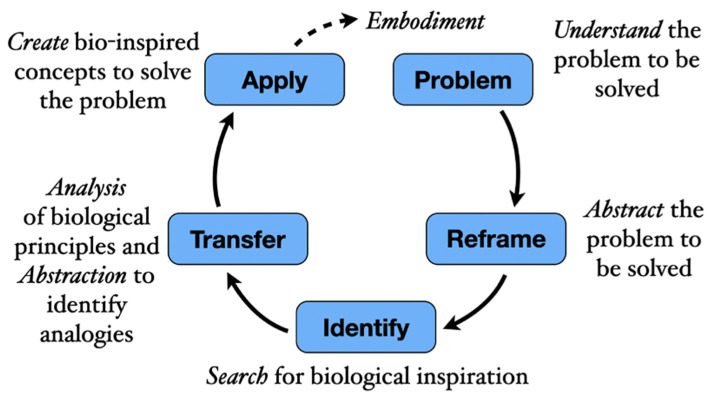
Fundamental BID process.

**Figure 2 biomimetics-07-00063-f002:**
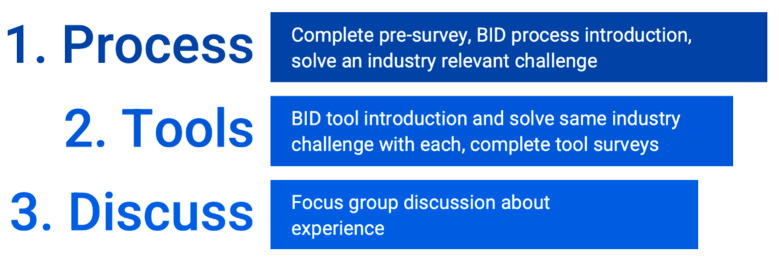
Training Overview.

**Figure 3 biomimetics-07-00063-f003:**
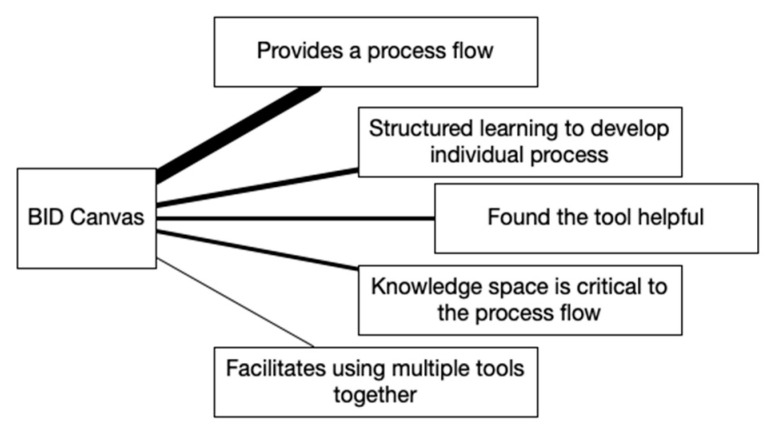
BID Canvas discussion themes.

**Figure 4 biomimetics-07-00063-f004:**
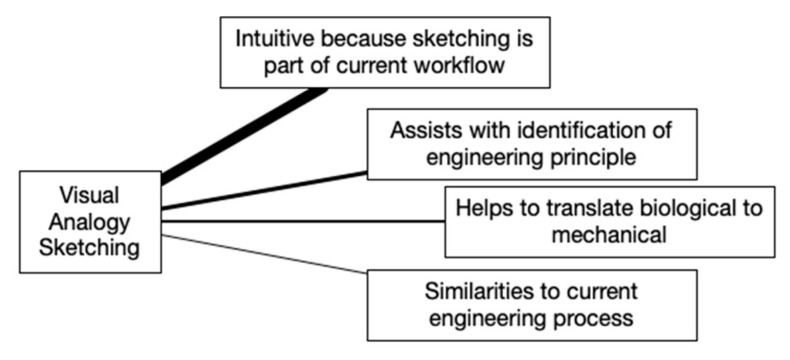
Visual Analogy Sketching discussion themes.

**Figure 5 biomimetics-07-00063-f005:**
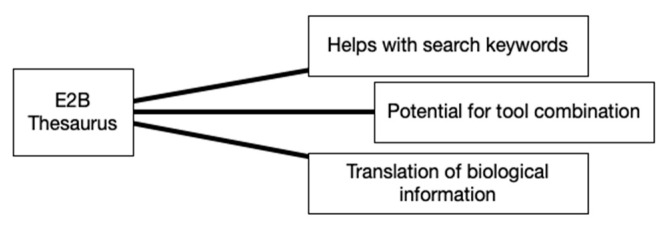
Engineering-to-Biology Thesaurus discussion themes.

**Figure 6 biomimetics-07-00063-f006:**
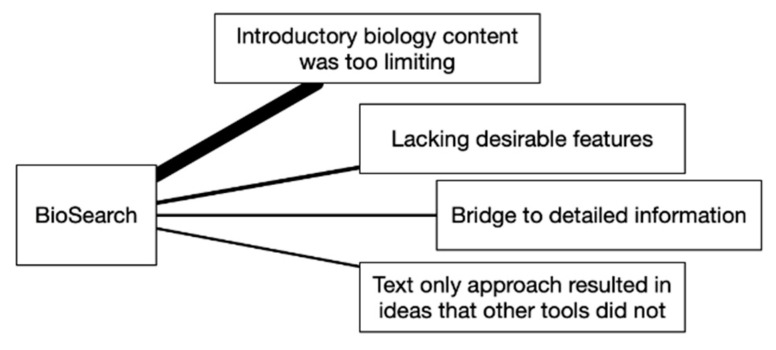
BioSearch discussion themes.

**Figure 7 biomimetics-07-00063-f007:**
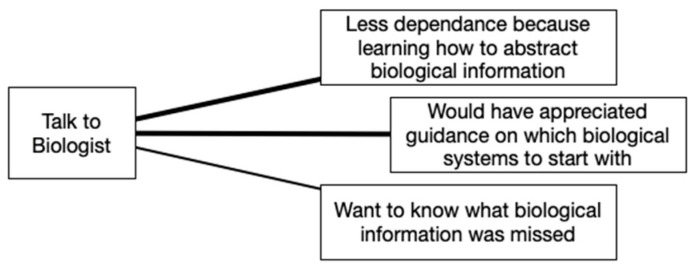
Talk to biologist discussion themes.

**Figure 8 biomimetics-07-00063-f008:**
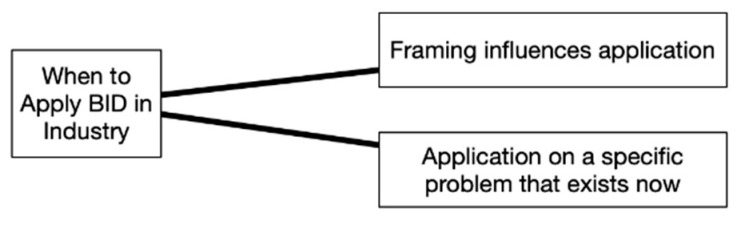
When to apply BID in industry discussion themes.

**Figure 9 biomimetics-07-00063-f009:**
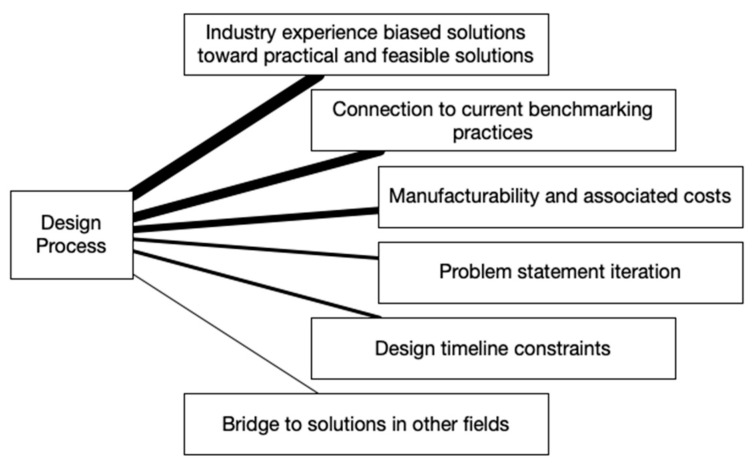
Design process discussion themes.

**Table 1 biomimetics-07-00063-t001:** Summary of BID Tools used in the training.

BID Tool	Description	BID Process Intention
Engineering-to-Biology Thesaurus	A tool that maps biology terms to engineering terms of the functional basis. Assists with translating biological information into an engineering context [32].	Reframe Identify Transfer
BioSearch	An application that supports searching a text only biology knowledge base using engineering terms. Assists with identification of suitable biological inspiration [55].	Identify Transfer
Visual Analogy Sketching	A technique that encourages sketching biological systems at varying magnification levels and detail to assist with developing analogies to engineering principles and components. Assists with thinking and reasoning about the form and function of the biological system [56].	Transfer Apply
BID Canvas	A visual guide based on concept-knowledge theory for structuring the thought patterns of the bio-inspired design process. Assists with scaffolding iterative connection-making [57,58].	Reframe Transfer Apply

**Table 2 biomimetics-07-00063-t002:** Participant roles and credentials breakdown.

Participant Current Role	Number of Participants in the Role	Educational Background of Participants in the Role
Associate Director of Advanced Development	2	B.S. Mechanical Engineering M.S. Applied Mathematics Ph.D. Engineering
Product Engineer	5	B.S. Mechanical Engineering B.S. Chemical Engineering B.S. Materials Science
Design Engineer	1	B.S. Mechanical Engineering M.F.A. Industrial Design
Operations Product Engineer	1	B.S. Mechanical Engineering
Industrial Designer	2	B.S. Industrial Design M.A. Industrial Design

**Table 3 biomimetics-07-00063-t003:** Quantitative results of BID tool feedback questions.

Question	Visual Analogy Sketching		BID Canvas		E2B Thesaurus		BioSearch	
	Mean	Mode	Mean	Mode	Mean	Mode	Mean	Mode
1. Rate your value of the tool you used	3.6	4	3.4	3	3.2	3	2.3	3
2. The tool effectively helped you work toward a BID solution	3.6	4	3.4	3	3.2	3	2.5	3
3. The tool improved your confidence about doing BID	3.6	4	3.1	3	2.7	3	2.2	2
4. The tool effectively helped you build connections between biology and the problem	3.1	4	3.0	3	3.0	3	2.8	3
5. The tool was intuitive enough to use on your own	3.6	4	2.7	3	3.2	3	2.7	3
6. The tool reduced your need/reliance on a biologist	3.0	4	2.6	2	2.7	2	2.2	3
7. Using the tool resulted in a better outcome than the baseline case (no BID tools)	3.0	3	2.7	3	2.3	2	2.5	3
8. You would use the tool again	3.6	4	3.6	4	3.0	3	2.3	1 and 3
**Total**	**27.0/32**		**24.6/32**		**23.2/32**		**19.5/32**	

**Table 4 biomimetics-07-00063-t004:** Expected and Unexpected BID Tool Use.

BID Tool	Expected Use	Unexpected Use
Engineering-to-Biology (E2B) Thesaurus	Reframe (5) Identify (4) Transfer (5)	Apply (2)
BioSearch	Identify (6)	Reframing (1) Transfer (2) Apply (1)
Visual Analogy Sketching	Transfer (7) Apply (6)	Problem (2) Reframe (1) Identify (4)
BID Canvas	Reframe (4) Transfer (6) Apply (6)	Problem (2) Identify (3)

## Data Availability

Not applicable.

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
