# Peer review of "Understanding the Use of Bio-Inspired Design Tools by Industry Professionals"

_biomimetics, 2022, doi:10.3390/biomimetics7020063_

Round 1

Reviewer 1 Report

I feel as though some really interesting information has been lost with the current analysis. These experts are not all the same. They have different experiences, educations, etc… and the analysis seems to lack this potential covariance. I found myself wondering whether the themes were divided up by participant type, who actually said each statement, and what we might learn about which type of expert would be best matched to which tool. And then, I read lines 566! LOL. That’s funny. You consider the participants to be similar and I don’t. That’s okay, for sure. But maybe consider where they might be different?

I also find it so incredibly interesting that your participants are referring to their ‘learning styles’. Learning styles are not a thing… there is no evidence to support that they actually exist. And yet, I am convinced, that the perception of having one has a HUGE influence throughout a person’s lifetime. The evidence for that is here too. Wow. I would so like to dive into this.

Line 158: Why was that specific challenge chosen?

Figure 2 is a bit jarring in its design and I’m also not quite sure what it is meant to convey.

Line 196: missing “the” at the beginning of the sentence.

Line 200: Change ‘the majority’ to ‘half’

Line 203: typo on ‘remining’.

Table 2 is cumbersome and difficult to read. Suggest redesign and reformatting.

Line 211: data ‘were’.

Line 221: can replace ‘and provides’ with ‘to provide’… eliminated repeated ‘provide’ in sentence.

Line 226: “these” data.

Line 227: “these” data.

Line 235: The Likert questions are uni directional. They are ‘leading’ in that there are no negative statements. Best practices would use some negative statements to ensure that the participant is paying attention to the survey questions, etc…

Line 271: ‘were’

Line 301: The use of averages can be deceiving. What about the distributions? Did they vary with type of participant? Were they different across evaluation items? A bimodal distribution will produce the same average as a normal distribution. I worry that some valuable information has been lost.

Line 303: table 4 is lovely

Lines 595 to 598 are absolutely critical messaging and, if at all possible, should be further emphasized. This is super important.

Line 620-626: I would argue that the remote nature of the experience mirrored ‘real life’ more authentically. These intense workshops are great but we all know that we don’t behave in the same way as if we were at home/work. I really think that the remote access is a strength of this work.

Reviewer 2 Report

What is the research question you seek to answer through this article?

The state of the art part is very brief; it is more of a case study that is presented.

What are the perspectives of this work?

Many references to specify!

Youtube.com: it's vast! Same for BBC, Forbes or TED!

Besides, there are too many citations from the author himself! (self-citations!) Other works exist!

Reviewer 3 Report

The paper is well written and easy to follow. It deals with a very important topic in the context of BID, i.e. how industry professionals can use BID tools during the design/development process. The paper gives valuable insight into training professionals and how to design tools so that they can be used easily on a daily basis in industry. As companies and product categories are so diverse, I was very curious how recommendations and generalizations could be deducted from the research. My expectations were completely met and it was a pleasure to read the paper. I have little comments and hope the authors consider them to be of value.

General
There are several double spaces at the beginning of sentences throughout the paper. 

Introduction

l. 48ff: I find the comparison of system properties between biology and engineering ("Plastic did not [...] and it is unlikely that a product will need to find a mate") a bit confusing, even though I see the point. It would be more professional to make a comparison that is reasonable, e.g. physical and chemical properties in both realms are similar, even though there are specificities which make comparisons more challenging, like biology evolves in a trial-and-error-process while engineering designs with a clear purpose. This raises the question how much of biology needs to be known in depth. As this is not the topic of the manuscript, references like Fish and Beneski. 2014, Evolution and Bio-Inspired Design: Natural Limitations, or Greaff et al. (2020). Biological practices and fields, missing pieces of the biomimetics’ methodological puzzle. Biomimetics, 5(4), 62, could be referred to.

l. 84ff: In Europe, OXO and Hydro Flask are not necessarily known. Maybe give a short description or link to these companies.

l. 86ff: "...it is important to note that no one company operates the same as another and [...]" This statement is true and makes any study so difficult. This is why I truly appreciate your work for this manuscript and that you share in detail what you found. However, this sentence questions in a way doing any study in this context. I am not sure what to do with this myself, and it raises the question how to customize or standardize the BID process - difficult to answer. Just a comment.

l. 98ff.: "Previous research on BID..." I agree mostly with your line of argumentation, but I was reminded of the work of Fayemi et al. 2017, Biomimetics: Process, tools and practice, and Graeff et al. 2021, Biomimetics from practical feedback to an interdisciplinary process, about testing the use of tools with practitioners; Chirazi et al. 2019, What do we learn from good practices, as well as Jacobs et al. 2014 BioM Innovation database, about developments in the field. I am sure, the authors are well aware of these references, so I was wondering why they were not included and based on their findings, some sentences are incomplete in their argumentation. No urgent need to include them, though.

Materials and methods

l. 158ff: If I understood correctly, all participants worked on the OXO topic. Did you notice any difference in working towards a BID between OXO-employees and non-OXO-employees?

table 1: Is reference [29] for the tool BioSearch correct? 

l. 196: What is meant with "Majority" here?

table 2: It would be interesting to know in which department the participants are working, e.g. is there an innovation department (with a lot of freedom in their work) or if participants work pre-product development or in the department of product development. 

l. 217 ff: Why is the creativity test necessary? What did it tell you? Did you see any connection with the outcome of the participants' work? What did you do with the results of this test?

Results

Table 3: It would be nice to add the total possible score, i.e. 32, the reader could immediately see that e.g. the Canvas scored 24.6 of 32. You could also ask the reader to remember that there was a scale of 1-4, but anyway, might me helpful.

Thank you again for your very nice work!!

Round 2

Reviewer 1 Report

The authors' response is absolutely satisfactory.  Thanks for the opportunity to review this work. I look forward to seeing it published.   

Reviewer 3 Report

Thank you again for the revisions, the paper looks really good!

One thing, that I forgot in my first review: in line 434 the "Nature4innovation site" is mentioned, but not explained anywhere else in the manuscript. Maybe not every reader knows what is meant here. Maybe the authors could briefly explain, what it is and where to find it.

The paper is ready for publication!